# Biosynthesized Silver Selenide Nanoparticles from *Meyerozyma guilliermondii* as a Novel Adjuvant to Revolutionize Gentamicin Therapy

**DOI:** 10.3390/microorganisms13122657

**Published:** 2025-11-22

**Authors:** Min Xu, Lei Yang, Ya-Wei Zhang, Chao Wu, Yuan-Yuan Cheng, Hao Xue

**Affiliations:** 1State Key Laboratory of Environmental Criteria and Risk Assessment, Chinese Research Academy of Environmental Sciences, Beijing 100012, China; renyumeiwen1987@163.com (M.X.); wuchao@ahu.edu.cn (C.W.); 2School of Life Sciences, Anhui University, Hefei 230601, China; d23301086@stu.ahu.edu.cn (L.Y.); chengyy@ahu.edu.cn (Y.-Y.C.); 3School of Resources and Environmental Engineering, Anhui University, Hefei 230601, China; zhangyw@163.com; 4Anhui Province Engineering Laboratory for Mine Ecological Remediation, Hefei 230601, China

**Keywords:** silver selenide nanoparticles, Ag_2_Se NPs, biosynthesis, antibiotics, synergy, gentamicin, biofilm, reactive oxygen species, membrane disruption

## Abstract

The increasing prevalence of antibiotic resistance necessitates the development of novel antimicrobial agents and therapeutic strategies. This study reports the extracellular biosynthesis of silver selenide nanoparticles (Ag_2_Se NPs) using *Meyerozyma guilliermondii* PG-1 and evaluates their antimicrobial and antibiofilm efficacy, both alone and in combination with gentamicin. The NPs were thoroughly characterized, confirming their nanoscale size, crystallinity, and biomolecule-mediated stability. Ag_2_Se NPs exhibited broad-spectrum antibacterial activity against Gram-positive (*Staphylococcus aureus*) and Gram-negative (*Pseudomonas aeruginosa*, *Escherichia coli*) pathogens and showed strong synergy with gentamicin, particularly against *P. aeruginosa* and *E. coli*, as demonstrated through checkerboard and time–kill assays. The NPs also significantly inhibited biofilm formation and disrupted pre-formed biofilms. Mechanistic studies revealed that the antibacterial effects involved membrane disruption, ATP leakage, and elevated oxidative stress, while gene expression analysis in *S. aureus* indicated triggered stress responses related to biofilm formation. These findings suggest that biosynthesized Ag_2_Se NPs represent a promising synergistic agent for enhancing antibiotic efficacy and combating biofilm-related infections.

## 1. Introduction

Antimicrobial resistance (AMR) continues to outpace the development of new antibiotics and undermines clinical management of high-burden pathogens such as *Pseudomonas aeruginosa* and *Staphylococcus aureus*, particularly in biofilm-associated infections where reduced penetration and stress-tolerant phenotypes limit drug efficacy [1,2]. Nanomaterials offer complementary solutions by combining intrinsic antibacterial activity with the capacity to potentiate existing antibiotics, notably aminoglycosides like gentamicin with various mechanisms [3,4,5]. Among these, silver-based nanostructures are prominent due to broad-spectrum activity and well-documented synergy with multiple antibiotics across Gram-negative and Gram-positive species, including clinical isolates [6,7]. Parallel advances in “green” or biogenic synthesis have produced colloidally stable, biomolecule-capped nanoparticles using microbial or plant systems, often improving dispersion and interfacial functionality relevant to bacterial surfaces while simplifying scale-up [8,9,10].

Despite this momentum, important gaps persist. First, while elemental Ag or Se nanoparticles (NPs) have been studied, there is limited evidence on biologically fabricated silver selenide (Ag_2_Se) NPs as antibiotic adjuvants. As a complex chalcogenide, Ag_2_Se could integrate Ag-associated membrane/redox insults with Se-associated bioactivity, potentially yielding distinct antibacterial and adjuvant profiles [5,11]. Second, species-specific mechanisms that link nanoparticle exposure to pharmacodynamic outcomes—synergy versus additivity—are not fully resolved. In particular, comparative analyses across *P. aeruginosa* and *S. aureus* that connect membrane integrity, ROS dynamics, ATP leakage, and transcriptional reprogramming of biofilm and lysis genes to checkerboard and time–kill readouts remain relatively scarce [2,12,13]. Third, although some studies report NPs–antibiotics potentiation, most of them rely on chemically synthesized particles or carrier systems. Pathway-informed microbial biosynthesis that yields well-defined, extracellularly produced nanocrystals with inherent capping and stability is less explored, yet attractive for translation [8,9,10].

Here, we establish an extracellular, green biosynthetic route using the yeast *Meyerozyma guilliermondii* PG-1 to produce well-dispersed, orthorhombic Ag_2_Se NPs and systematically evaluate their antibacterial performance alone and as adjuvants to gentamicin against reference strains of *P. aeruginosa*, *Escherichia coli*, and *S. aureus*. We show strong synergy against *P. aeruginosa* and *E. coli* and an additive, non-antagonistic interaction against *S. aureus* in checkerboard assays, which is mirrored by accelerated and durable killing in time–kill kinetics. We further demonstrate potent antibiofilm effects in both prevention and disruption settings. Mechanistically, we integrate multi-angle evidence—morphology and membrane integrity, ROS induction, ATP leakage, and targeted transcriptional shifts in *S. aureus* (*icaA*, *cidA/lrgA*, *agrA*, *hla*, *psmA*)—to develop a species-resolved framework that rationalizes why catastrophic envelope failure in Gram-negative bacteria favors strong synergy, whereas a defensive EPS/eDNA program in *S. aureus* tempers synergy under static MIC conditions yet permits profound bactericidal outcomes dynamically. This work aligns with and extends the emerging paradigm of nano-enabled antibiotic revitalization by highlighting actionable levers—membrane access, ROS dynamics, and biofilm gene regulation—to guide rational NPs–antibiotics co-therapy design and future translational development [6,14].

## 2. Materials and Methods

### 2.1. Strains and Maintenance

The yeast strain *M.*
*guilliermondii* PG-1 used in this study was originally isolated and identified from soil samples collected in the Daqing Oil Field, Heilongjiang Province, China. This strain has been deposited in the China Center for Type Culture Collection (CCTCC) with the accession number CCTCC AY 2020004.

For antimicrobial assays, the following strains were employed: *E.*
*coli* ATCC 25922, *P.*
*aeruginosa* ATCC 27853, and *S*. *aureus* ATCC 29213. All strains were routinely cultivated in Lysogeny Broth (LB) medium containing 10 g/L tryptone, 5 g/L yeast extract, and 10 g/L NaCl, unless stated otherwise. All culture medium reagents were obtained from Sangon Biotech Co., Ltd. (Shanghai, China).

### 2.2. Biosynthesis and Purification of Ag_2_Se Nanoparticles

For the biosynthesis of Ag_2_Se NPs, an overnight culture of *M. guilliermondii* PG-1 grown in LB medium was harvested by centrifugation at 8000× *g* for 5 min. The cell pellet of *M. guilliermondii* PG-1 was washed three times with sterile R2A medium and resuspended in fresh R2A medium containing the following components (per liter): 0.5 g tryptone, 0.5 g yeast extract, 0.5 g casamino acids, 0.5 g soluble starch, 0.05 g MgSO_4_·7H_2_O, 13.42 g K_2_HPO_4_, and 11.80 g anhydrous sodium citrate. The cell suspension was inoculated into R2A medium supplemented with 5.55 mM glucose and 1.5 mM sodium pyruvate to an initial optical density at 600 nm (OD_600_) of 0.1. After 24 h of incubation at 30 °C with shaking at 180 rpm (late-log phase), AgNO_3_ and Na_2_SeO_3_ were added to the culture to a final concentration of 1 mM each. The incubation continued for another 48 h under the same conditions. The culture broth was centrifuged (8000× *g*, 10 min) to remove cells, and the supernatant was collected. The supernatant was dialyzed using a 1 kDa molecular weight cut-off membrane against deionized water for 24 h and then filtered through a 0.22 μm membrane to obtain purified Ag_2_Se nanoparticles. The resulting nanoparticle suspension was stored at 4 °C prior to further characterization and use.

To evaluate the biosynthetic capability of different cellular fractions, a 24-h culture of *M. guilliermondii* PG-1 in R2A medium was centrifuged to separate the cells from the cell-free supernatant. The supernatant was divided into two aliquots: one was filtered through a 0.22 μm sterile membrane, and the other was sterilized by autoclaving (121 °C, 20 min). The cell pellet was resuspended in R2A medium and disrupted by ultrasonication on ice, followed by filtration through a 0.22 μm membrane to remove cellular debris. All three samples—filtered supernatant, heat-sterilized supernatant, and filtered cell lysate—were adjusted to equal volumes, supplemented with 1 mM AgNO_3_ and 1 mM Na_2_SeO_3_, and incubated for 48 h. After incubation, the mixtures were centrifuged, and the absorbance of the supernatants was measured at 260 nm to compare the relative efficiency of nanoparticle synthesis. Na_2_SeO_3_ and AgNO_3_ were purchased from China National Pharmaceutical Group Corporation (Beijing, China). All other chemicals and medium components were obtained from Sangon Biotech Co., Ltd. (Shanghai, China).

In addition, a metabolic inhibition assay was conducted. Sodium molybdate (Na_2_MoO_4_) and sodium tungstate (Na_2_WO_4_) were prepared as sterile stock solutions. Late-log *M. guilliermondii* PG-1 cultures in R2A were used as described above. Inhibitors were added to a final concentration of 20 mM and cultures were pre-incubated for 30 min at 30 °C with shaking. AgNO_3_ and Na_2_SeO_3_ (1 mM each, final) were then added and incubation was continued for 48 h under the same conditions. Post-incubation processing and A_260_ measurement followed the procedure described for the culture-fraction assay. Relative biosynthetic activity (%) was calculated as (A_260_ of inhibitor-treated culture/A_260_ of untreated intact-culture control) × 100, with the untreated intact culture defined as 100%. Experiments were performed with *n* = 4 biological replicates.

### 2.3. Characterization of Ag_2_Se Nanoparticle

Total elemental silver (Ag) in the purified Ag_2_Se NPs suspension was quantified by inductively coupled plasma mass spectrometry (ICP-MS, Thermo Fisher Scientific, Waltham, MA, USA). Briefly, 1 mL of the purified Ag_2_Se NPs suspension was transferred into a digestion vessel and mixed with 12 mL of freshly prepared aqua regia (concentrated HCl:HNO_3_ = 3:1, *v*/*v*). The mixture was heated on a hotplate at 180 °C until the volume was reduced to approximately 1 mL. Heating was then stopped and the digest was brought to volume and serially diluted to a final 10^5^-fold dilution in a volumetric flask. The Ag concentration in the diluted solution was determined by ICP-MS using external calibration with Ag standards; procedural blanks were included and subtracted. The morphology of the nanoparticles before and after purification was observed using a JEM-F200 transmission electron microscope (JEOL, Tokyo, Japan). Elemental composition was analyzed by energy-dispersive X-ray spectroscopy (EDS) equipped on the TEM, operating at an acceleration voltage of 200 kV. The size distribution of the purified nanoparticles was measured based on TEM images using ImageJ software (version 1.54g, National Institutes of Health, Bethesda, MD, USA). Crystalline structure was examined by high-resolution transmission electron microscopy (HR-TEM), from which lattice fringe spacing and selected-area electron diffraction (SAED) patterns were obtained. The surface charge (zeta potential) of nanoparticles in aqueous suspension was measured using a Zetasizer Nano ZS instrument (Malvern Panalytical, Malvern, UK). For structural and chemical characterization, purified Ag_2_Se NPs were lyophilized and gently ground into powder. Crystal structure was analyzed by X-ray diffraction (XRD; Rigaku, Tokyo, Japan). Phase identification and peak indexing were conducted by matching the experimental diffraction peaks to the reference pattern for orthorhombic Ag_2_Se (PDF#24-1041) from the International Centre for Diffraction Data (ICDD) database, confirming the orthorhombic phase assignment. Surface elemental composition and chemical states were determined by X-ray photoelectron spectroscopy (XPS; JEOL, Tokyo, Japan). Surface functional groups were identified using Fourier transform infrared spectroscopy (FT-IR; Bruker, Karlsruhe, Germany). The hydrodynamic size distribution of the biosynthesized Ag_2_Se NPs was determined using dynamic light scattering (DLS) (Zetasizer Nano ZS90, Malvern Instruments Ltd., Malvern, UK). Samples were transferred into a disposable polystyrene cuvette, equilibrated at 25 °C for 2 min, and analyzed at a scattering angle of 90°. The intensity-weighted particle size distribution was recorded as the mean of three independent measurements. For stability evaluation, freshly purified Ag_2_Se NPs and samples stored at 4 °C for two months were measured under identical conditions.

### 2.4. Antimicrobial Susceptibility Testing

#### 2.4.1. Determination of Minimum Inhibitory Concentration (MIC)

The antimicrobial activity of biosynthesized Ag_2_Se NPs was evaluated against *S. aureus*, *E. coli*, and *P. aeruginosa* following the Clinical and Laboratory Standards Institute (CLSI) guidelines with appropriate modifications for nanoparticle testing [1,2,3]. All experiments were performed in Mueller–Hinton Broth (MHB). Overnight cultures were adjusted to a turbidity of a 0.5 McFarland standard (approximately 1 × 10^8^ CFU/mL) using sterile saline (0.85% NaCl). This standardized suspension was then diluted in fresh MHB to achieve a working inoculum of approximately 5 × 10^5^ CFU/mL. Two-fold serial dilutions of Ag_2_Se NPs and/or gentamicin were prepared in sterile 96-well plates using MHB as the diluent. Subsequently, 100 μL of the prepared working inoculum was added to each well, resulting in a final test volume of 200 μL per well and a final bacterial density of approximately 2.5 × 10^5^ CFU/mL. The plates were incubated statically at 37 °C for 20 to 24 h. After incubation, the MIC was defined as the lowest concentration that completely inhibited visible growth. All assays were performed in triplicate.

#### 2.4.2. Checkerboard Assay for Synergy Evaluation (FICI Calculation)

Synergistic effects between Ag_2_Se NPs and gentamicin were evaluated using the checkerboard microdilution method [15]. Two-dimensional arrays were prepared in 96-well plates with serial two-fold dilutions of Ag_2_Se NPs along the *x*-axis and gentamicin along the *y*-axis. The final concentrations ranged from 0.0625 to 64 mg/L for NPs and 0.125 to 8 mg/L for gentamicin. Each well was inoculated with bacterial suspension to achieve a final density of approximately 2.5 × 10^5^ CFU/mL. After incubation at 37 °C for 20 h, the fractional inhibitory concentration index (FICI) was calculated using the formula: FICI = (MIC of drug A in combination/MIC of drug A alone) + (MIC of drug B in combination/MIC of drug B alone). The interaction was interpreted as follows: synergy (FICI ≤ 0.5), additive (0.75 < FICI ≤ 1.0), indifferent (1.0 < FICI ≤ 4.0), or antagonism (FICI > 4.0). All assays were performed in triplicate.

#### 2.4.3. Time–Kill Kinetics Assay

Time–kill kinetics were determined for *S. aureus* ATCC 29213, *E. coli* ATCC 25922, and *P. aeruginosa* ATCC 27853 according to established procedures with minor modifications [15]. Overnight cultures were adjusted to a 0.5 McFarland standard in sterile saline and diluted in MHB to obtain a starting inoculum of approximately 1 × 10^6^ CFU/mL. Test groups included Ag_2_Se NPs alone at 1× and 4× their predetermined MIC, gentamicin alone at 1× and 4× MIC, and the combination of Ag_2_Se NPs and gentamicin with each agent at 1× MIC. A growth control (MHB without antimicrobial agents) was included for each strain. Cultures were incubated at 37 °C with agitation at 200 rpm. At 0, 2, 4, 8, and 24 h, samples were withdrawn, immediately subjected to serial 10-fold dilutions in sterile saline to minimize antimicrobial carryover, and aliquots from appropriate dilutions were spread onto Mueller–Hinton Agar (MHA) plates. Plates were incubated at 37 °C for 18–24 h, and colony-forming units (CFU) were enumerated and expressed as log_10_ CFU/mL.

Bactericidal activity was defined as 99.9% reduction in CFU/mL relative to the initial inoculum at time zero, following CLSI M26-A recommendations [15]. Synergy for the combination treatment was defined as 99% decrease in CFU/mL at 24 h compared with the most active single agent tested alone [5,15,16]. All experiments were performed in triplicate biological replicates.

### 2.5. Antibiofilm Assays

The anti-biofilm efficacy of Ag_2_Se NPs was evaluated against *S. aureus* and *P. aeruginosa* using two complementary approaches: a biofilm prevention assay to determine the ability to inhibit biofilm formation and a biofilm disruption assay to assess the eradication activity against pre-formed mature biofilms, following established methods with modifications [17]. For the biofilm prevention assay, overnight cultures were diluted to approximately 1 × 10^6^ CFU/mL in fresh Tryptic Soy Broth (TSB) supplemented with 1% glucose. Aliquots of this bacterial culture were added to individual wells of a sterile 96-well plate. Subsequently, Ag_2_Se NPs suspension was immediately added to the wells to achieve desired final concentrations. Wells containing bacterial culture with 0.1 M phosphate-buffer saline (PBS, pH 7.4) served as the positive control for biofilm growth, while wells containing only medium served as the negative control. The plates were then incubated statically for 24 h at 37 °C. Following the respective incubation periods for each assay, the treatment agents were aspirated and all wells were gently washed twice with PBS to remove non-adherent cells. The anti-biofilm effects of NPs were quantified by assessing the total adhered biomass of the residual biofilm. The total biomass was evaluated using a crystal violet staining protocol, wherein biofilms were fixed with methanol, stained with 0.1% (*w*/*v*) crystal violet solution, and solubilized with 33% glacial acetic acid for optical density measurement at 570 nm.

For the biofilm disruption assay, mature biofilms were first established by inoculating 200 μL of the diluted bacterial subculture into 96-well plates and incubating statically for 24 h at 37 °C. After the formation period, the supernatant was carefully aspirated and the biofilms were gently washed twice with PBS. Then, 180 μL of fresh TSB was added to each well, followed by the addition of 20 μL of Ag_2_Se NPs suspension to achieve the desired final concentrations. Wells containing only fresh medium were used as the untreated biofilm control. The plates were further incubated statically for an additional 24 h at 37 °C. Results are presented as relative biofilm biomass, with the untreated biofilm control normalized to 100%: relative biomass (%) = (OD_570_treated/OD_570_control) × 100.

### 2.6. Mechanistic Studies of Antimicrobial Action

#### 2.6.1. Scanning Electron Microscopy (SEM) Sample Preparation

Subcultures of *E. coli* and *S. aureus* ATCC were treated with Ag_2_Se NPs at 1× MIC for 2 h, following a similar process to the killing assay described above. After treatment, cells were collected by centrifugation at 12,000× *g* for 1 min, washed twice with PBS containing 1.25% glutaraldehyde, and incubated at 4 °C for fixation. After 2 h, the cells were collected again by centrifugation and dehydrated using ethanol solutions at various concentrations (35%, 50%, 75%, and 100%). Cells were treated for 10 min with ethanol solution at each concentration. The cells in 100% ethanol were vigorously suspended immediately before being dropped onto silicon slices for SEM observation.

#### 2.6.2. Membrane Integrity Assay

Subcultures of *E. coli* and *S. aureus* ATCC were treated with Ag_2_Se NPs at 1× MIC for 2 h, following a similar process to the killing assay described above. After treatment, the cells were collected, washed twice with sterile saline, and stained with SYTO 9 and propidium iodide (PI) from the LIVE/DEAD BacLight Bacterial Viability Kit (Thermo Fisher Scientific Inc., Waltham, MA, USA), following the manufacturer’s instructions. The live cells exhibited green fluorescence, while the dead cells exhibited red fluorescence. The images of live and dead cells were captured using an Inverted fluorescence microscope (Leica, Wetzlar, Germany).

#### 2.6.3. Measurement of Intracellular Reactive Oxygen Species (ROS)

Overnight cultures of *S. aureus* and *E. coli* were diluted in fresh Mueller–Hinton (MH) broth and grown to early logarithmic phase. Cells were harvested, washed, and resuspended in phosphate-buffered saline (PBS) to approximately 1 × 10^6^ CFU/mL. The bacterial suspension was incubated with 2′,7′-dichlorodihydrofluorescein diacetate (DCFH-DA) at a final concentration of 20 μM for 20 min at 37 °C in the dark. After incubation, cells were gently centrifuged and washed twice with PBS to remove unincorporated dye, then resuspended in PBS to the original volume. The cell suspension was aliquoted into black 96-well plates and immediately treated with Ag_2_Se NPs at 0, 1/4×, 1/2×, 1×, and 4× the MIC. Plates were incubated at 37 °C for 2 h. The antioxidant inhibition experiment was conducted using *N*-acetyl-L-cysteine (NAC) as a reactive oxygen species scavenger. Early-logarithmic cultures of *E. coli* ATCC 25922 and *S. aureus* ATCC 29213 were pre-loaded with 20 µM DCFH-DA probe as described above and subsequently treated with Ag_2_Se NPs at 1/2× MIC for 2 h at 37 °C, in the presence of increasing NAC concentrations (0, 1 and 5 mM). A group without NPs and NAC was included as the no-NPs control. Fluorescence was measured with excitation/emission at 488/525 nm using a multimode microplate reader. Results were expressed as fluorescence intensity relative to the untreated control. Experiments were performed in triplicate.

#### 2.6.4. ATP Leakage Assay

The leakage of intracellular adenosine triphosphate (ATP) was assessed as an indicator of membrane damage. *E. coli* ATCC 25922 and *S. aureus* ATCC 29213 in early logarithmic phase were harvested, washed twice with PBS, and resuspended in PBS to an OD_600_ of 1. Bacterial cultures were treated with Ag_2_Se NPs at final concentrations of 1/2×, 1×, 4×, and 10× MIC for 2 h at 37 °C. Following treatment, samples were immediately centrifuged at 12,000× *g* for 10 min at 4 °C to pellet cells and debris, and the supernatant was collected. For negative controls, bacteria cultures were treated with PBS alone. For positive controls (maximum ATP release), cells were subjected to complete lysis. *E. coli* c suspensions were disrupted by ultrasonication on ice. *S. aureus* suspensions were incubated with lysostaphin (20 μg/mL) at 37 °C for 30 min and then ultrasonicated on ice. Lysates were centrifuged as above to obtain supernatants containing the total releasable ATP. ATP concentrations in supernatants were determined using a commercial luciferin–luciferase ATP assay kit (Beyotime, Shanghai, China), following the manufacturer’s instructions.

#### 2.6.5. RNA Extraction and Quantitative Real-Time PCR (qPCR)

Overnight cultures of *S. aureus* ATCC 29213 were diluted in fresh MHB to an initial OD_600_ of 0.01 and incubated at 37 °C with shaking at 200 rpm until mid-log phase. At that point, cultures were split such that one portion received Ag2Se NPs at 1× MIC while the other received an equivalent volume of PBS as the untreated control, and incubation was continued for 1 h at 37 °C. Immediately thereafter, cells were pelleted by centrifugation (12,000× *g*, 1 min, 4 °C) and promptly lysed/stabilized by adding NucleoZOL (Takara, Beijing, China). Lysates were stored at −80 °C until extraction. For total RNA isolation, samples were processed following the manufacturer’s instructions. Total RNA was further cleaned using an RNA clean-up kit (BioTeke, Beijing, China) with on-column DNase I digestion as recommended. RNA yield and purity were examined on a NanoDrop spectrophotometer (Thermo Fisher Scientific) and by agarose gel electrophoresis showing distinct rRNA bands. Genomic DNA removal and cDNA synthesis was conducted a kit HiScript III All-in-one RT SuperMix Perfect for qPCR (Vazyme International LLC., Nanjing, China) according to the manufacturer’s instructions. Thereafter, qRT-PCR was performed with the StepOne real-time PCR system (Applied Biosystems, Foster City, CA, USA) using synthesized cDNA and SupRealQ Ultra Hunter SYBR qPCR Master Mix(U+) (Vazyme International LLC., Nanjing, China). Relative expression values of target genes were obtained from three determinations with normalization against 16S rRNA using the method of 2^−ΔΔCt^.

### 2.7. Statistics

Data are presented as the mean ± standard deviation (SD) of at least three independent biological replicates. Statistical analysis was performed using one-way ANOVA with Tukey’s post hoc test. A *p*-value of less than 0.05 was considered statistically significant (*p* < 0.05). All analyses were performed using GraphPad Prism version 9.0.0.

## 3. Results

### 3.1. Biosynthesis and Multifaceted Characterization of Meyerozyma-Derived Ag_2_Se Nanoparticles

The extracellular biosynthesis of nanoparticles (NPs) in the culture supernatant of *M. guilliermondii* PG-1 was first confirmed by UV–Vis spectroscopy. As shown in Figure 1a, the spectrum of the purified product displays a distinct, broad absorption peak centered at approximately 260 nm. This feature is characteristic of silver selenide nanostructures and is consistent with successful Ag_2_Se NPs formation.

HR-TEM was used to assess morphology, size distribution, and crystallinity. The HR-TEM images (Figure 1b) show predominantly spherical NPs. Clear lattice fringes with interplanar spacings of 0.33 nm and 0.26 nm correspond to the (111) and (112) planes of orthorhombic Ag_2_Se, respectively [18]. The particle size distribution (Figure 1c) indicates a narrow distribution with an average diameter of 5.8 ± 2.1 nm, indicating effective stabilization during biosynthesis. Energy-dispersive X-ray spectroscopy (EDS) (Figure 1d) confirms the presence of silver and selenium as the main elemental components. X-ray diffraction (XRD) analysis (Figure 1e) further identifies the orthorhombic Ag_2_Se phase, with diffraction peaks at 26.7°, 33.4°, 36.7°, and 43.4°, indexed to the (111), (112), (004), and (201) planes (PDF#24-1041), respectively.

Colloidal stability was evaluated by zeta potential measurement, yielding −9.47 mV (Figure 1f). This negative surface charge suggests stabilization via electrostatic repulsion [19] and indicates the presence of anionic biomolecules capping the NPs surface. Fourier-transform infrared (FTIR) spectroscopy (Figure 1g) supports this interpretation, revealing characteristic absorption bands: a broad band at 3423 cm^−1^ (O-H stretching), a band at 2932 cm^−1^ (C-H stretching), peaks at 1652 cm^−1^ and 1396 cm^−1^ (assigned to Ag–Se-related vibrations) (104), a band at 1249 cm^−1^ (C-O stretching), and a band at 1084 cm^−1^ (C-O and P-O stretching in peptidoglycan). The presence of functional groups such as hydroxyl, carboxyl, and phosphate is consistent with the observed negative zeta potential, as these groups can deprotonate in aqueous media and contribute to surface charge. Collectively, the FTIR results indicate that the Ag_2_Se NPs are capped by biomolecules (e.g., lipids, carbohydrates, proteins), which contribute to structural stability and prevent aggregation. To assess the reproducibility and time-dependent stability of the biosynthesized Ag_2_Se nanoparticles, dynamic light scattering (DLS) analysis was conducted on freshly purified samples and those stored at 4 °C for two months. The freshly prepared Ag_2_Se NPs displayed a hydrodynamic size peak at 12.2 nm, while the stored suspension retained a similar unimodal distribution with a peak at 14.2 nm (Figure 1h). The slight decrease in hydrodynamic diameter indicates the preservation of colloidal dispersion and surface integrity, consistent with the stabilizing effect of the biomolecular capping derived from *M. guilliermondii*. These DLS observations verify that the biosynthetic route yields Ag_2_Se NPs with good stability and batch-to-batch reproducibility, supporting their reliability for downstream biological applications.

### 3.2. Extracellular Secretions Are Primarily Responsible for Ag_2_Se NPs Biosynthesis

To elucidate the mechanism of extracellular biosynthesis of Ag_2_Se NPs, the relative efficiency of NPs biosynthesis across different culture fractions was assessed by measuring the characteristic absorption at 260 nm (Figure 2a). A late-log culture of *M. guilliermondii* PG-1 was separated into three aliquots: one retained as intact culture for direct NP synthesis, one centrifuged to obtain cell-free supernatant, and one subjected to ultrasonication to prepare cell lysate. The intact culture demonstrated the highest activity and was set as the reference (100%). The cell-free supernatant retained a significant portion of the biosynthetic capability, exhibiting 60.6 ± 3.4% relative activity, whereas the cell lysate showed reduced efficacy with only 19.9 ± 1.3%. These results indicate that the primary reducing agents responsible for NPs synthesis are actively secreted extracellularly rather than predominantly associated with intracellular components.

To further investigate the nature of these extracellular reducing agents, we hypothesized that their biosynthesis and secretion depend on specific metabolic pathways. Sulfate assimilation and reduction, which are crucial for generating cellular reducing equivalents, were considered a candidate pathway. To test this hypothesis, specific metabolic inhibitors targeting this pathway were employed. The addition of sodium molybdate, a competitive inhibitor of sulfate transport and sulfate-metabolizing enzymes, led to a pronounced inhibition of NPs formation, reflected by an ~81% decrease to 19.1 ± 2.3% relative activity compared with the uninhibited control (Figure 2b). In contrast, sodium tungstate, a less effective analogue, had a negligible effect, yielding 99.6 ± 3.9% relative activity, similar to the control (Figure 2b). The stark difference between molybdate and tungstate, two chemically similar oxyanions, supports a pathway-specific inhibition consistent with disruption of sulfate metabolism rather than nonspecific metal toxicity. Taken together, these data indicate pathway-specific inhibition consistent with disruption of sulfate metabolism rather than nonspecific metal toxicity. For clarity, all nanoparticles employed in subsequent antibacterial and antibiofilm assays were produced extracellularly from intact (non-lysed) cultures and recovered from clarified, cell-free supernatants, then purified as described in Section 2.2. Unless otherwise specified, this purified extracellular preparation was used for all biological experiments, whereas other fractions were analyzed only to trace nanoparticle formation and localization, not for activity testing.

### 3.3. Antimicrobial Performance of Ag_2_Se NPs Alone and Combination of Gentamicin

Following the successful synthesis and characterization of Ag_2_Se NPs, we evaluated their antimicrobial properties. The intrinsic antibacterial activity against clinically relevant reference strains was quantified by determining MIC, with gentamicin as a benchmark comparator. Ag_2_Se NPs exhibited broad-spectrum activity, inhibiting both Gram-negative (*P. aeruginosa*, *E. coli*) and Gram-positive (*S. aureus*) pathogens (Table 1). The NPs showed potent activity against *P. aeruginosa* ATCC 27853, with MIC comparable to gentamicin. Activity against *S. aureus* ATCC 29213 was similarly strong, while *E. coli* ATCC 25922 was the most susceptible, showing the lowest MIC among the tested strains.

Given aminoglycoside resistance challenges, particularly in Gram-negative bacteria, we assessed potential synergy between Ag_2_Se NPs and gentamicin using a checkerboard microdilution assay. Checkerboard assays revealed strong synergy between Ag_2_Se NPs and gentamicin against *P. aeruginosa* ATCC 27853 with FICI = 0.25 (Table 1). In combination, MICs for the agents were reduced relative to their individual MICs (Table 1). Synergy was also observed against *E. coli* ATCC 25922 with FICI = 0.31 (Table 1). No synergy was observed against *S. aureus* ATCC 29213; the interaction was additive (FICI = 0.62) without evidence of antagonism (Table 1).

To evaluate dynamic antibacterial efficacy, time–kill assays were conducted against *P. aeruginosa*, *E. coli*, and *S. aureus* over 24 h. Against *P. aeruginosa*, gentamicin at 1× MIC produced an initial bactericidal effect followed by regrowth, consistent with adaptive resistance and heteroresistance under sub-lethal pressure. In contrast, the combination yielded a rapid, sustained bactericidal effect, and no viable bacteria were detected from 8 h onwards (Figure 3a). Against *E. coli*, gentamicin at 4× MIC reached a bactericidal endpoint (<LOD) by 8 h, whereas the combination at 1× MIC for each agent accelerated killing, achieving the same endpoint by 4 h and sustaining it through 24 h (Figure 3b). Ag_2_Se NPs alone at 4× MIC showed slower killing than gentamicin and failed to eradicate the population. Against *S. aureus*, all monotherapies were bacteriostatic. Gentamicin at 4× MIC reduced bacterial counts by approximately 99.9% but did not eradicate the population at 24 h. Ag_2_Se NPs alone at 4× MIC showed similar inhibition without a bactericidal endpoint. In contrast, the combination at 1× MIC for each agent achieved a rapid and profound bactericidal effect, reducing viable counts by >99.99% (to 0.5 log_10_ CFU/mL) by 8 h (Figure 3c). Although a minimal regrowth to 0.9 log_10_ CFU/mL was observed at 24 h, this still represents a >99.99% reduction in viable bacteria compared to the control, indicating a sustained bactericidal outcome without evidence of antagonism, consistent with the additive interaction observed by FICI.

We next assessed the antibiofilm activity of Ag_2_Se NPs against *P. aeruginosa* and *S. aureus*, evaluating their ability to both prevent new biofilm formation and disrupt established biofilms. Ag_2_Se NPs inhibited biofilm formation in a concentration-dependent manner in both strains. For *P. aeruginosa*, 1/2× MIC reduced biofilm biomass to 14.9% of the control (85.1% inhibition) (Figure 4a). A more potent effect was observed for *S. aureus*, where 1/2× MIC reduced biofilm biomass to 6.7% of the control (93.3% inhibition) (Figure 4c). Ag_2_Se NPs also disrupted established biofilms. Against mature *P. aeruginosa* biofilms, 4× MIC reduced biomass by 91.2% (Figure 4b). Treatment of mature *S. aureus* biofilms with 4× MIC resulted in a 76% reduction in biomass (Figure 4d). For both strains, biofilm disruption was dose-dependent. At concentrations of 1× MIC and below, biofilm biomass was reduced by approximately 40–50%.

### 3.4. Mechanistic Insights into Antibacterial Activity of Ag_2_Se NPs

Having established the exceptional synergistic antibacterial and anti-biofilm efficacy of Ag_2_Se NPs, we next sought to delineate the underlying mechanisms of action. A multi-faceted approach was employed to investigate the physical and biochemical interactions between the nanoparticles and bacterial cells.

#### 3.4.1. Ag_2_Se NP Induce Species-Specific Membrane Disruption and Stress Responses

To probe the dynamic antibacterial mechanisms of Ag_2_Se NPs, we investigated membrane damage at a critical sub-lethal time point, as informed by our time–kill kinetics data. This condition was strategically selected because it represents a pivotal stage where a substantial proportion of the bacterial population undergoes active stress and damage—yielding clear analytical signals—while retaining sufficient cell viability for robust assessment. SEM imaging provided direct visual evidence of membrane damage and revealed intriguing species-specific responses to Ag_2_Se NPs treatment. In contrast to the smooth surfaces of untreated cells, NPs-treated *E. coli* displayed a shriveled morphology with apparent perforations (Figure 5a,b), a classic sign of membrane disruption that compromises its integrity and barrier function. In contrast, the response in *S. aureus* was markedly different. Untreated cells exhibited characteristically smooth and intact surfaces (Figure 5c). In contrast, cells exposed to NPs were extensively covered with a dense, flocculent mat of extracellular material. This substance exhibited a fibrous, web-like morphology, creating filamentous bridges that entangled individual cells into clusters (Figure 5d). This is a highly indicative finding, as it suggests that the NPs may not only attack the membrane but also trigger a robust stress response, potentially stimulating the exaggerated secretion of exopolysaccharides or other components of the extracellular polymeric substance (EPS) in a defensive, albeit ultimately futile, reaction.

The loss of membrane integrity was further confirmed using a dual-fluorescence viability stain (SYTO 9 and PI). SYTO 9 labels all cells (green), while propidium iodide (PI) only penetrates cells with compromised membranes, quenching SYTO 9 fluorescence and staining them red. In untreated control groups, the vast majority of both *E. coli* (Figure 5e,i) and *S. aureus* (Figure 5g,k) cells exhibited bright green fluorescence with virtually no red fluorescence. This strong SYTO 9 uptake in the absence of PI staining confirms that the cells were viable with intact membranes. However, after treatment with Ag_2_Se NPs, a substantial increase in red fluorescence was accompanied by a concomitant decrease in green fluorescence. This shift in fluorescence was nearly complete in *E. coli* (Figure 5f,j), indicating widespread loss of membrane integrity. The effect was less pronounced in *S. aureus* (Figure 5h,l), where a higher proportion of cells retained green fluorescence, suggesting a differential response to NP-induced damage.

#### 3.4.2. Ag_2_Se NPs Trigger a Cascade of Intracellular Oxidative Stress

To determine whether membrane disruption caused by Ag_2_Se NPs was accompanied by intracellular oxidative stress, we measured ROS generation in both *E. coli* and *S. aureus* following exposure to sub-inhibitory and inhibitory concentrations of Ag_2_Se NPs. Exposure time is a key determinant of both phenotypic and molecular readouts. The time points were selected a priori based on time–kill kinetics. At a concentration of 4× MIC, both *E. coli* and *S. aureus* exhibited significant loss of viability within 2 h. We therefore used a 2-h exposure for phenotypic assays that report acute stress and membrane damage—specifically, ROS production and ATP leakage—because these readouts are most discernible at a strongly bactericidal yet quantifiable time point. The results revealed a potent and concentration-dependent induction of ROS in both species, albeit with distinct patterns.

In *E. coli*, a dramatic escalation in ROS levels was observed (Figure 6a), peaking at a greater than 6-fold increase over the baseline at the 1× MIC treatment. However, a further increase in NPs concentration to 4× MIC resulted in a slight attenuation of the ROS signal, although it remained highly elevated at approximately 5.7-fold. This attenuation suggests the onset of an extremely rapid cytotoxic effect at lethally high concentrations, potentially leading to a metabolic collapse that precedes or curtails the full measurable oxidative burst. In contrast, *S. aureus* exhibited a more pronounced and linear dose–response (Figure 6b), with ROS levels increasing relentlessly across the entire concentration gradient to reach a striking ~13-fold increase at 4× MIC. This sustained response implies a different mode of interaction, possibly facilitated by the thick cell wall sequestering NPs and enabling a prolonged, cumulative oxidative assault on intracellular targets. These findings establish that a rapid oxidative burst is a key component of the antibacterial mechanism of Ag_2_Se NPs, acting in concert with the direct physical membrane damage previously documented. The stark difference in ROS kinetics between the two species highlights a fundamentally different path to cell death: a potentially saturable and catastrophic breakdown in *E. coli* versus a relentless and additive oxidative escalation in *S. aureus*. To further confirm that ROS generation contributed to the antibacterial effect of Ag_2_Se NPs, we performed an NAC rescue experiment in which bacterial cells were co-treated with the antioxidant *N*-acetyl-L-cysteine (NAC). As shown in Figure 6c,d, the addition of NAC effectively alleviated NP-induced bacterial killing. For *E. coli*, co-incubation with 1 mM NAC completely restored bacterial viability, indicating that the cell death induced by 1/2× MIC Ag_2_Se NPs was fully rescued at this NAC concentration. Increasing NAC to 5 mM did not further enhance the protective effect, yielding a survival level comparable to that of 1 mM (Figure 6c). In contrast, for *S. aureus*, 1 mM NAC only partially rescued bacterial viability, whereas treatment with 5 mM NAC achieved complete protection, restoring the cell survival rate to that of the untreated control (Figure 6d). These differential responses suggest that *S. aureus* may experience more persistent oxidative stress than *E. coli* under the same NPs exposure conditions. Collectively, these results indicate that, under the low-dose condition (1/2× MIC), the bactericidal effect of Ag_2_Se NPs is closely associated with ROS generation, suggesting that oxidative stress plays a major role in the antibacterial activity at sub-inhibitory concentrations.

#### 3.4.3. Ag_2_Se NPs Cause Leakage of ATP

To further investigate the membrane disruption observed in the previous section, we evaluated whether Ag_2_Se NPs induced the leakage of intracellular ATP in both bacterial species. The results demonstrated a clear concentration-dependent increase in extracellular ATP upon treatment with Ag_2_Se NPs, though the extent of leakage varied markedly between *E. coli* and *S. aureus*.

In *E. coli* (Figure 7b), even a sub-inhibitory concentration (1/2× MIC) of Ag_2_Se NPs resulted in detectable ATP leakage, with extracellular levels reaching approximately 12% of the positive lysed-cell control. As the NPs concentration increased, this effect became more pronounced: at 1× MIC, leakage rose to about 30%, and at 4× MIC, nearly 75% of the total cellular ATP was released. Notably, at 10× MIC, the extracellular ATP level slightly exceeded that of the lysed-cell control, suggesting almost complete disruption of membrane integrity—possibly even more efficient than the mechanical lysis method used for the control. In contrast, *S. aureus* exhibited a more resistant profile (Figure 7b). At 1/2× MIC, ATP leakage was minimal, accounting for only about 5% of the positive control value. Although higher NPs concentrations led to increased ATP release—reaching roughly 17% at 1× MIC, 36% at 4× MIC, and 72% at 10× MIC—even the highest concentration did not result in complete ATP release compared to the enzymatically lysed control. These findings reinforce the species-specific antibacterial action of Ag_2_Se NPs. The substantial ATP leakage observed in *E. coli* supports the earlier morphological evidence of severe membrane damage, while the comparatively limited leakage in *S. aureus* aligns with its reduced susceptibility and suggests a more resilient cell envelope structure. Together, these results indicate that membrane permeabilization and consequent loss of cytoplasmic components represent a key mechanism of Ag_2_Se NPs toxicity.

#### 3.4.4. Ag_2_Se NPs Alter the Transcription of Biofilm-Related Genes in *S. aureus*

The observation of a dense, flocculent extracellular matrix in SEM analysis prompted a molecular investigation to determine its composition and origin. To elucidate whether this material represented a targeted bacterial stress response, we analyzed the expression of key genes involved in the production of major extracellular polymeric substance (EPS) components. In contrast to measurements of ROS production and ATP release, such extensive killing is suboptimal for transcript analysis: RNA extracted after approximately 90% cell death is highly heterogeneous due to ongoing lysis and degradation, which can bias gene expression profiles. To capture an early and biologically meaningful transcriptional response while minimizing this confounding effect, we shortened the exposure period to 1 h for qPCR, focusing on *S. aureus* genes associated with biofilm formation. The qPCR analysis revealed a highly coordinated transcriptional shift in *S. aureus* following 1-h exposure to 1× MIC Ag_2_Se NPs. A 1.8-fold upregulation of *icaA* (Figure 8), a critical gene for the synthesis of polysaccharide intercellular adhesin (PIA), was detected, indicating a robust effort by the bacterium to produce exopolysaccharides. Concurrently, a dysregulation was observed in the *cid*/*lrg* system, which governs controlled cell lysis and the release of extracellular DNA (eDNA). Expression of the pro-lytic gene *cidA* was increased (1.68-fold), while the anti-lytic gene *lrgA* was significantly suppressed (0.57-fold) (Figure 8). This imbalance probably creates a strong net drive towards cell wall degradation and lysis, directly accounting for the leakage of intracellular ATP quantified in the previous section and the fibrous, web-like structures that suggest the presence of stretched genomic DNA within the EPS matrix. The expression of genes encoding protein-based virulence factors, namely *hla* (alpha-toxin) and *psmA* (phenol-soluble modulin), was reduced (Figure 8). Notably, the central quorum-sensing regulator *agrA*, typically orchestrates a transition towards an invasive lifestyle, showed a slight downregulation (Figure 8). The suppression of the agr system suggests a strategic trade-off; under severe membrane stress inflicted by the NPs, the bacterium appears to divert resources away from energy-costly toxin production and instead prioritizes the emergency synthesis of protective EPS and the release of eDNA for biofilm scaffolding.

## 4. Discussion

This study establishes the first reported extracellular biosynthesis of silver selenide nanoparticles (Ag_2_Se NPs) using *Meyerozyma guilliermondii*, demonstrating their potential as potent adjuvants for gentamicin therapy against clinically relevant bacterial pathogens. The integration of green nanotechnology with antimicrobial synergy represents a paradigm shift toward sustainable approaches for combating multidrug-resistant infections, particularly given the current stagnation in traditional antibiotic development.

The successful extracellular production of Ag_2_Se NPs by *M. guilliermondii* PG-1 extends the established capacity of this yeast species to synthesize noble metal nanoparticles. Our findings align with previous research documenting that *M. guilliermondii* can synthesize silver and gold nanoparticles through both intracellular and extracellular pathways, but our work represents the first documentation of Ag_2_Se NPs biosynthesis by this organism. The distinctive UV-vis absorption peak at 260 nm, combined with XRD confirmation of the orthorhombic phase [18], validates the formation of crystalline Ag_2_Se rather than elemental components. Our mechanistic investigation reveals that extracellular secretions account for approximately 60% of the biosynthetic capacity, contrasting with some yeast-mediated synthesis systems where intracellular reduction predominates. The dramatic inhibition by sodium molybdate (~81% reduction) implicates sulfate assimilation and reduction pathways [8], suggesting an enzymatically driven mechanism rather than nonspecific metal complexation. This biogenic approach offers significant advantages over traditional chemical synthesis methods, which typically require high temperatures or toxic reducing agents [20,21].

The antimicrobial performance reveals fascinating species-specific patterns that illuminate the complex interplay between nanoparticle-induced stress and bacterial adaptive responses. The strong synergy against Gram-negative pathogens (FICI = 0.25 for *P. aeruginosa*, 0.31 for *E. coli*) contrasts with the additive interaction in *S. aureus* (FICI = 0.62), yet all combinations achieved profound bactericidal effects in time–kill assays. This apparent contradiction between static synergy metrics and dynamic killing outcomes highlights the limitations of traditional checkerboard assays and underscores the value of mechanistic investigation [20,21,22,23]. Our results demonstrate that Ag_2_Se NPs induced substantial ROS accumulation within bacterial cells, as evidenced by DCFH-DA fluorescence assays. The validation using N-acetyl-L-cysteine (NAC) provides direct evidence that oxidative stress contributes causally to the antibacterial mechanism. The differential NAC requirements for rescue (1 mM for *E. coli*, 5 mM for *S. aureus*) suggest that Gram-positive bacteria experience more persistent oxidative stress, possibly due to their thicker peptidoglycan layer sequestering nanoparticles and enabling prolonged ROS generation. This mechanism parallels reports of AgNPs inducing cellular damage through ROS production [22], but extends these findings to the more complex chalcogenide system of Ag_2_Se.

The membrane disruption patterns revealed by SEM and fluorescence staining illuminate why synergy manifests differently across species. In *E. coli*, catastrophic envelope failure with visible perforations creates direct pathways for gentamicin uptake, explaining the rapid sterilization observed in time–kill assays. This mechanism aligns with previous studies showing that silver-based nanoparticles can disrupt bacterial membranes and enhance antibiotic penetration [23]. The massive ATP leakage approaching 75% at moderate concentrations further supports complete membrane compromise in Gram-negative bacteria. Conversely, the complex defensive response mounted by *S. aureus* provides mechanistic insight into why static synergy is weaker despite achieving excellent bactericidal outcomes. The dense, web-like extracellular matrix visualized by SEM, combined with coordinated transcriptional reprogramming, reveals a sophisticated stress response program. The upregulation of *icaA* (1.8-fold) indicates accelerated polysaccharide intercellular adhesin synthesis, while the imbalanced *cidA*/*lrgA* expression suggests a shift toward controlled autolysis and enhanced eDNA release [24,25,26]. This response initially creates barriers that may impede gentamicin penetration in static assays, but the metabolic stress and structural vulnerabilities ultimately render cells more susceptible to combination therapy.

The potent antibiofilm activity observed in both prevention and disruption assays (>90% reduction in *P. aeruginosa*, >75% in *S. aureus*) has significant clinical implications, as biofilm-associated infections are notoriously difficult to treat due to reduced drug penetration and stress-tolerant phenotypes [1,2]. The enhanced biofilm disruption likely results from the same multi-modal mechanism observed in planktonic cultures: direct matrix interaction, ROS-mediated damage, and facilitated antibiotic penetration. The *S. aureus* transcriptional signature provides additional insight, as the increased polysaccharide production and eDNA release may initially strengthen the matrix but ultimately destabilize it through uncontrolled autolysis. Similar “penetrate and collapse” strategies using nanoparticle–antibiotic combinations have been documented for various biofilm-forming pathogens [24,27,28].

The choice of Ag_2_Se NPs over conventional AgNPs offers theoretical advantages that merit further investigation. Selenium compounds possess well-documented antioxidant and anti-inflammatory properties [29], which could theoretically provide protective effects against gentamicin-associated nephrotoxicity [30]. This dual functionality—antimicrobial silver combined with potentially protective selenium—represents a novel approach to mitigating antibiotic side effects while enhancing efficacy. The concept of using biosynthesized nanoparticles as adjuvants in gentamicin therapy is well-supported by existing research, where AgNPs have demonstrated the ability to combat biofilms and enhance bacterial eradication while potentially mitigating required dosages [31]. Furthermore, biosynthesized AgNPs have shown anti-nephrotoxic effects against gentamicin-induced toxicity in animal models [30], suggesting that Ag_2_Se NPs could potentially provide similar protective benefits.

However, several important questions remain for clinical translation. The stability and speciation behavior of Ag_2_Se NPs in complex biological matrices requires systematic investigation, as protein corona formation and ion release can dramatically alter nanoparticle behavior [32]. Comprehensive in vivo pharmacokinetics, biodistribution, and long-term toxicological profiles are essential. Resistance development under prolonged sub-inhibitory exposure also requires attention [4]. The integration of such biogenic nanoparticles as adjuvants in clinical settings involves careful consideration of their stability, biocompatibility, and long-term toxicity profiles, especially given examples like Prussian Blue nanoparticles that exhibit pH-dependent degradation [32].

The integration of biosynthesized Ag_2_Se NPs into clinical practice will likely require sophisticated delivery platforms to maximize therapeutic index while minimizing systemic exposure. Liposomal formulations, hydrogel matrices, and targeted nanocarriers have successfully concentrated antibiotics at infection sites while reducing off-target effects [24,33,34]. The application of silver nanoparticles in advanced wound dressings, such as hydrogels and microneedle patches, demonstrates the potential for localized delivery approaches that could enhance healing while preventing infections [35,36]. Incorporating Ag_2_Se NPs into such platforms could enable localized delivery for high-burden applications such as chronic wounds, orthopedic implants, or pulmonary infections.

In conclusion, this work demonstrates that biogenically synthesized Ag_2_Se NPs represent a promising platform for revitalizing gentamicin therapy through multi-modal antibacterial mechanisms involving membrane disruption, oxidative stress, and enhanced antibiotic penetration. The species-specific responses revealed here provide actionable insights for rational design of nanoparticle–antibiotic combinations, highlighting the importance of considering bacterial defensive programming in therapeutic development. Future research should focus on comprehensive safety assessment, clinical formulation development, and systematic evaluation of resistance trajectories to realize the translational potential of this innovative approach to antimicrobial therapy.

## Figures and Tables

**Figure 1 microorganisms-13-02657-f001:**
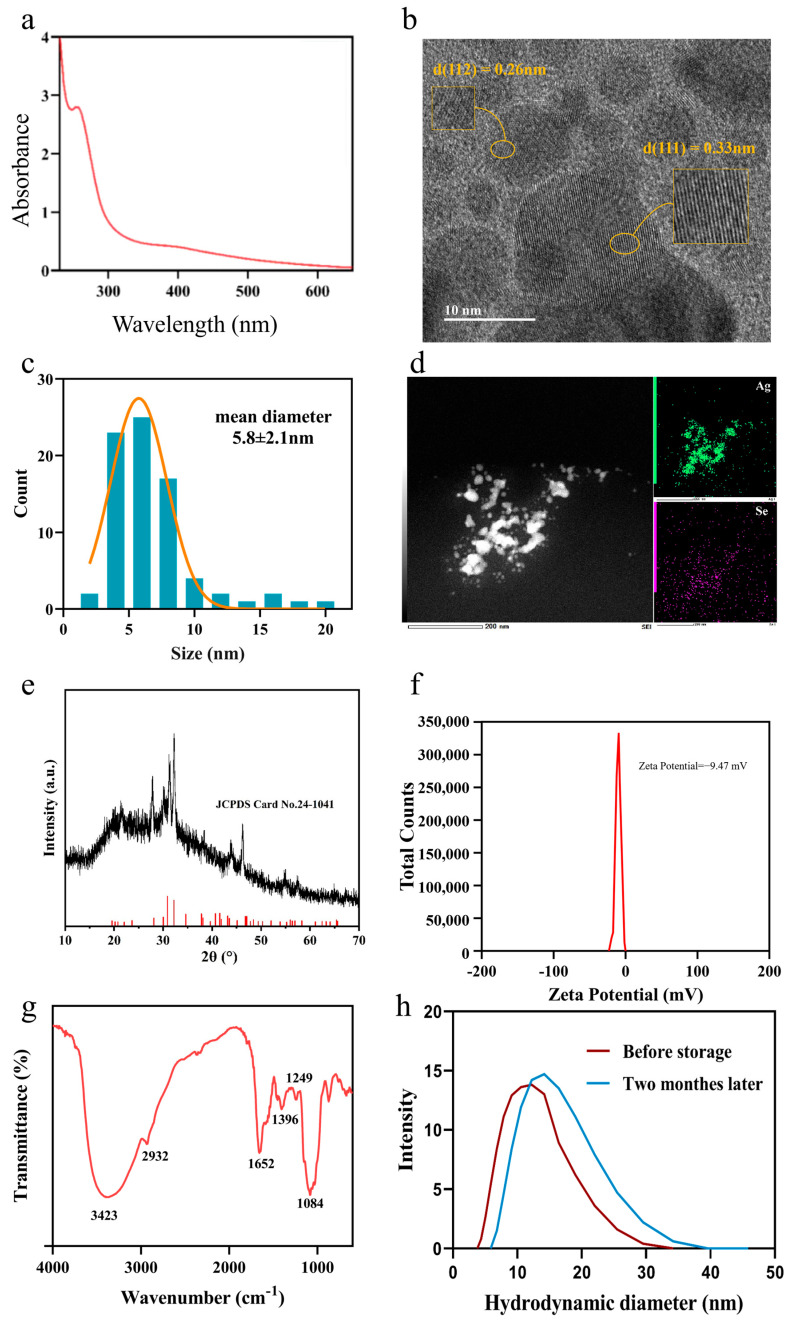
Biosynthesis and characterization of extracellular Ag_2_Se nanoparticles produced by *M. guilliermondii* PG-1. (**a**) UV–Vis absorption spectrum of the purified nanoparticles. (**b**) HR–TEM micrograph of the nanoparticles after synthesis. (**c**) Particle size distribution histogram derived from TEM measurements. (**d**) Energy-dispersive X-ray spectroscopy (EDS) spectrum, (**e**) X-ray diffraction (XRD) patternof the purified Ag_2_Se NPs (black) and the standard reference pattern for Ag_2_Se (JCPDS card No. 24-1041, red). (**f**) Zeta potential distribution, and (**g**) Fourier-transform infrared (FTIR) of purified nanoparticles. (**h**) DLS size distribution of biosynthesized Ag_2_Se nanoparticles freshly prepared and after two-month storage at 4 °C.

**Figure 2 microorganisms-13-02657-f002:**
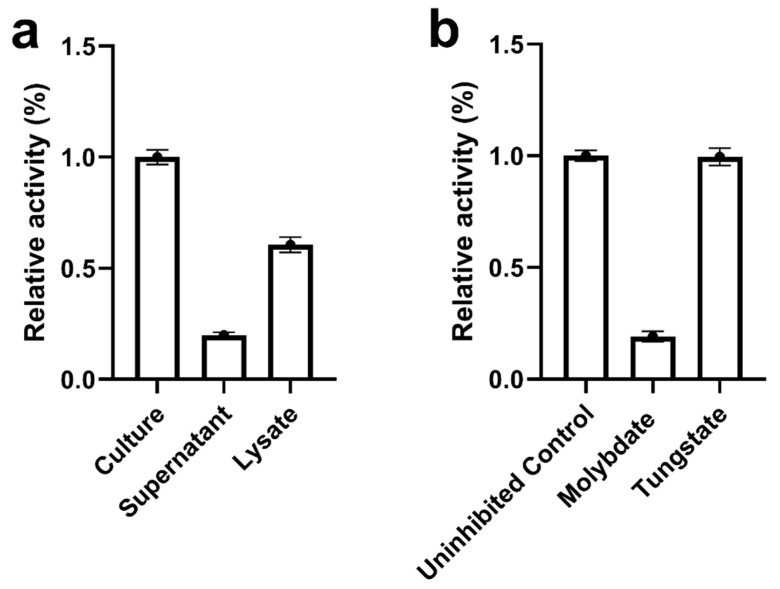
Analysis of Ag_2_Se NPs biosynthetic capacity. (**a**) Relative activity (%) of different *M. guilliermondii* PG-1 culture fractions (intact culture, cell-free supernatant, cell lysate) in Ag_2_Se NPs biosynthesis; the intact culture was set to 100%. (**b**) Relative activity (%) of intact culture in Ag_2_Se NPs biosynthesis following treatment with sodium molybdate or sodium tungstate; the untreated control was set to 100%. Data are presented as mean ± SD (*n* = 4).

**Figure 3 microorganisms-13-02657-f003:**
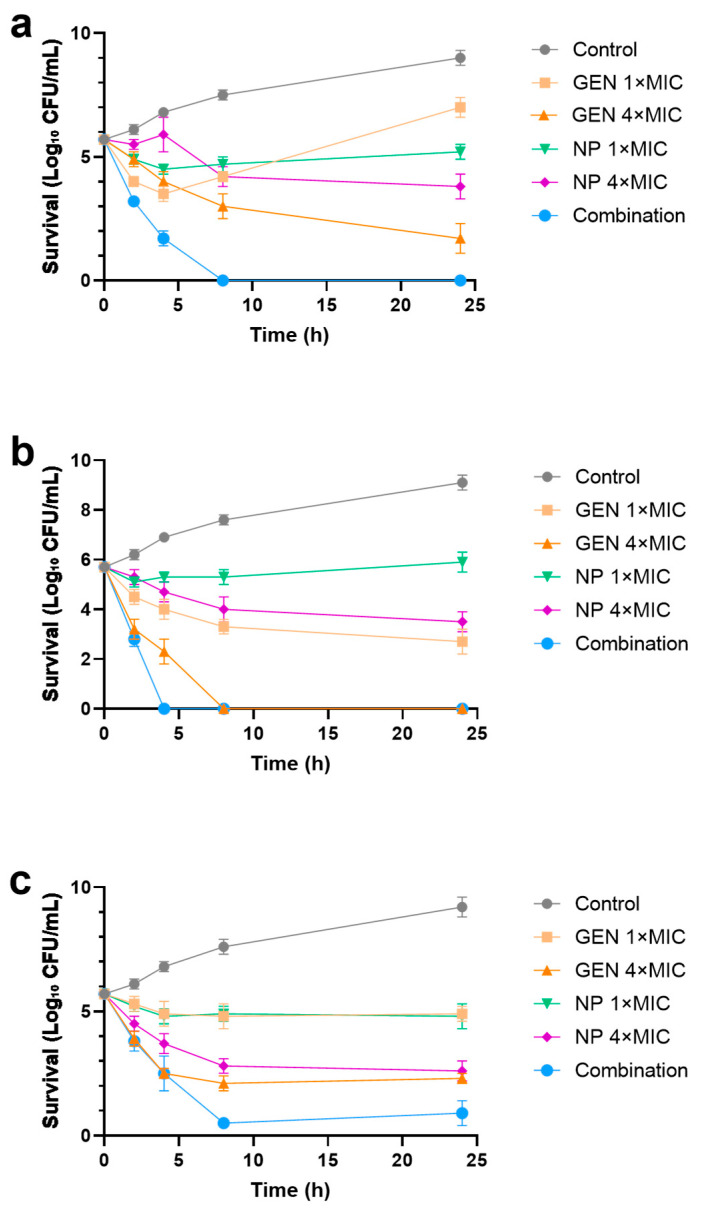
Time–kill kinetics of Ag_2_Se NPs, gentamicin (GEN), and their combination against (**a**) *P. aeruginosa*, (**b**) *E. coli*, and (**c**) *S. aureus*. Bacterial cultures were treated with Ag_2_Se NPs alone, GEN alone, or their combination at the specified multiples of the MIC for each agent. Viability was determined by measuring survival (Log_10_ CFU/mL) over 24 h. Data are presented as mean ± SD (*n* = 3).

**Figure 4 microorganisms-13-02657-f004:**
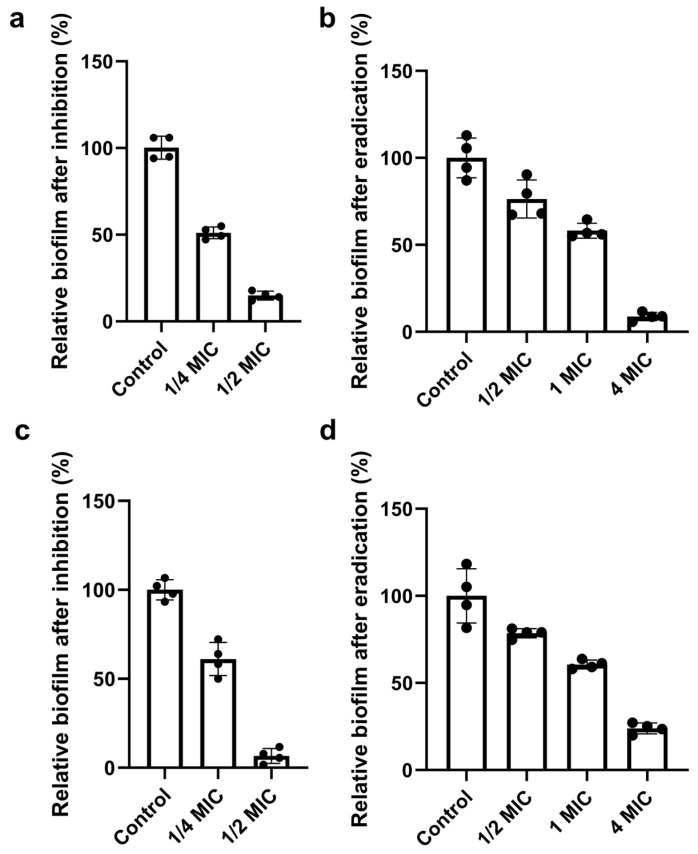
Concentration-dependent antibiofilm activity of Ag_2_Se NPs against *P. aeruginosa* and *S. aureus*. (**a**) Prevention of *P. aeruginosa* biofilm formation. Biofilm biomass after 24 h incubation with sub-MIC concentrations of Ag_2_Se NPs, expressed as a percentage of the untreated control. (**b**) Disruption of pre-established *P. aeruginosa* biofilms. Mature biofilms were treated with Ag_2_Se NPs at the indicated concentrations for 24 h. (**c**) Prevention of *S. aureus* biofilm formation. (**d**) Disruption of pre-established *S. aureus* biofilms. Data are presented as mean ± SD (*n* = 4).

**Figure 5 microorganisms-13-02657-f005:**
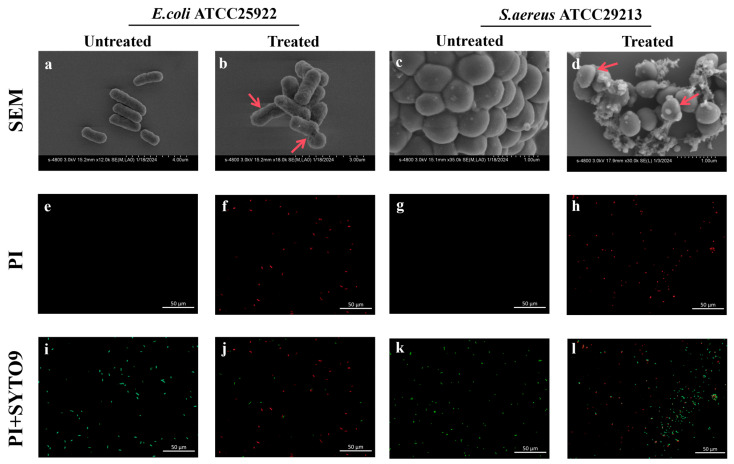
Assessment of bacterial morphology and membrane integrity following treatment with 1× MIC Ag_2_Se NPs for 2 h. (**a**–**d**) Scanning electron microscopy (SEM) images of untreated (**a**,**c**) and Ag_2_Se NPs-treated (**b**,**d**) *E. coli* (**a**,**b**) and *S. aureus* (**c**,**d**). Red arrows in (a–d) mark areas of cellular damage in the SEM images. (**e**–**h**) Fluorescence microscopy images of bacterial cells stained with propidium iodide (PI). Untreated (**e**,**g**) and Ag_2_Se NPs-treated (**f**,**h**) *E. coli* (**e**,**f**) and *S. aureus* (**g**,**h**). (**i**–**l**) Fluorescence microscopy images of bacterial cells co-stained with SYTO 9 and PI. Untreated (**i**,**k**) and Ag_2_Se NPs-treated (**j**,**l**) *E. coli* (**i**,**j**) and *S. aureus* (**k**,**l**).

**Figure 6 microorganisms-13-02657-f006:**
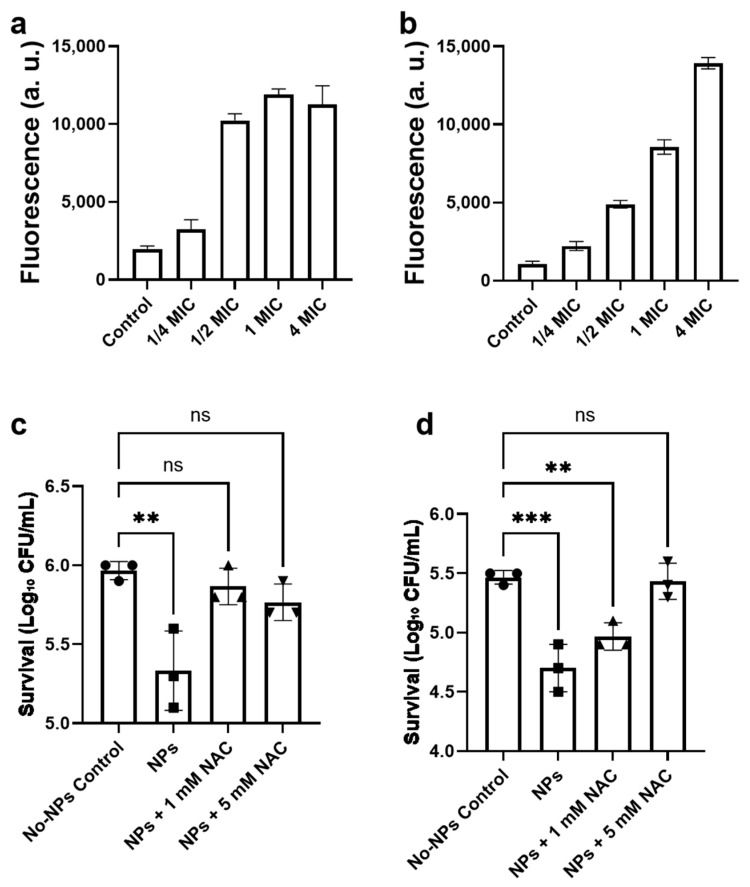
ROS generation and antioxidant inhibition in *E. coli* and *S. aureus* exposed to Ag_2_Se NPs. Intracellular ROS fluorescence intensity in (**a**) *E. coli* and (**b**) *S. aureus* after treatment with Ag_2_Se NPs at indicated multiples of the MIC for 2 h. Suppression of Ag_2_Se NP-induced ROS by N-acetyl-L-cysteine (NAC) in (**c**) *E. coli* and (**d**) *S. aureus*. Cells were treated with1/2× MIC Ag_2_Se NPs for 2 h in the presence of 0, 1, or 5 mM NAC; “No-NPs control” denotes cells without NPs or NAC. Data are expressed as mean ± SD (*n* = 3). Statistical significance was evaluated using one-way ANOVA followed by Tukey’s multiple comparison test (******* *p* < 0.01, ****** *p* < 0.01; ns, not significant *p* ≥ 0.05).

**Figure 7 microorganisms-13-02657-f007:**
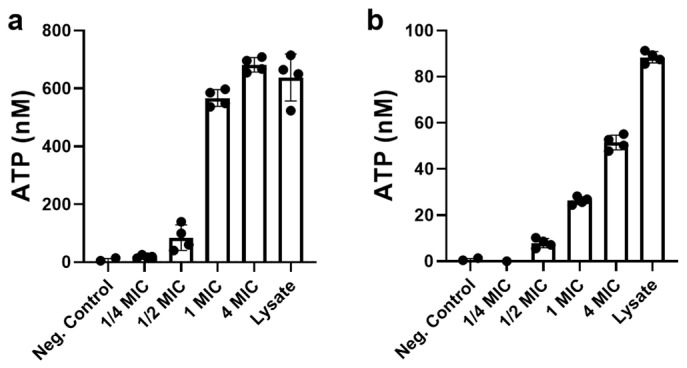
ATP leaked from cells of (**a**) *E. coli* and (**b**) *S. aureus* exposed to Ag_2_Se NPs. Bacterial suspensions were treated with Ag_2_Se NPs and the amount of ATP leaked from cells into the supernatant was quantified, where the value for the lysed cell control was defined as 100%. Neg Control represents untreated bacterial cells (negative control). Data are presented as mean ± SD (*n* = 4).

**Figure 8 microorganisms-13-02657-f008:**
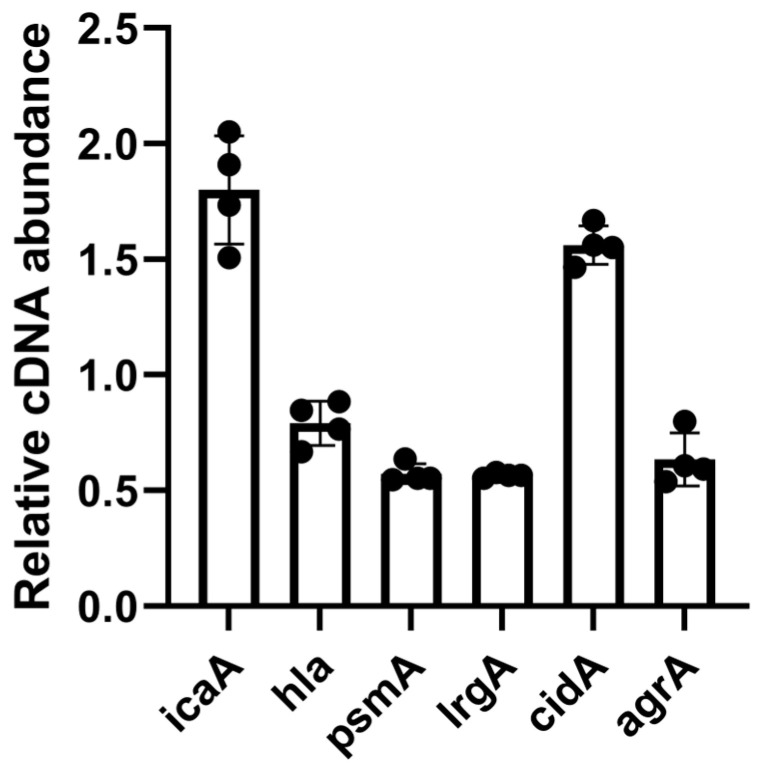
Ag_2_Se NPs alter the transcription of biofilm-related genes in *S. aureus*. Cultures were exposed to 1× MIC Ag_2_Se NPs for 1 h. Gene expression was quantified by qPCR and is presented as relative cDNA abundance of the treated group compared to the untreated control. Data show the mean ± SD of biological replicates.

**Table 1 microorganisms-13-02657-t001:** Minimum Inhibitory Concentration (MIC) and Fractional Inhibitory Concentration Index (FICI) values (mg/L) of Ag_2_Se NPs and gentamicin against reference strains.

Strains	Ag_2_Se	Gentamicin	FICI	Interaction
*P. aeruginosa* ATCC 27853	0.5	0.5	0.25	Synergy
*E. coli* ATCC 25922	0.25	0.5	0.31	Synergy
*S. aureus* ATCC 29213	0.5	0.5	0.62	Additive

## Data Availability

The raw data supporting the conclusions of this article will be made available by the authors on request.

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
