# Peer review of "Biosynthesized Silver Selenide Nanoparticles from Meyerozyma guilliermondii as a Novel Adjuvant to Revolutionize Gentamicin Therapy"

_microorganisms, 2025, doi:10.3390/microorganisms13122657_

Round 1
Reviewer 1 Report
Comments and Suggestions for Authors
The manuscript written by Min Xu et al. is a comprehensive study of the biosynthesis of Ag₂Se NPs using Meyerozyma guilliermondii. Their antibacterial activity in combination with gentamicin was assessed. The manuscript is well written, the NPs obtained were thoroughly characterized using appropriate techniques, and the topic is relevant to antimicrobial resistance and nanomaterial-assisted potentiation of antibiotics.
My comments are the following:
- Row 94 – the following is grammatically incorrect: “The cell pellet was washed three times with…. and resuspended in R2A medium containing the following components (per liter)”. The authors should explain what the cell pellet was washed with. “Washed three times with R2A medium and resuspended in fresh R2A medium”
- Fig. 1e-The authors should check the XRD pattern and JCPDS card representation. In the current representation, the JCPDS card doesn’t match the corresponding diffraction lines.
- While the manuscript provides SEM images of planktonic bacteria treated with Ag₂Se NPs (Figure 5), no SEM images that demonstrate the effect of the nanoparticles or the Ag₂Se + gentamicin combination on biofilm structures are provided. Since antibiofilm efficacy is a significant claim of the study, the manuscript would be substantially strengthened by including SEM images of biofilms before and after treatment.
Reviewer 2 Report
Comments and Suggestions for Authors
The authors present a method for the synthesis of Ag₂Se nanoparticles through a green approach using yeast.
The manuscript is overall complete and of great interest, with well-described and complementary analyses that clearly outline the effects of these nanoparticles on different bacterial strains.
Nevertheless, the work could be further improved by making some modifications to the text and the presented figures.
In the Materials and Methods section, the authors should better clarify how they calculated the concentration of the nanoparticles. Is the concentration expressed in mg/mL referring to the absolute weight or to one of the two main elements?
Furthermore, the authors should explain somewhere in the text why ROS, PCR, and other analyses were performed 2 hours after exposure to the nanoparticles. What is the rationale behind this choice? The time of expore could affect the results obtained in the different assays.
As a general consideration, figure magnification or dimensions should be improved. For example, in Figure 1, the EDS panel is not very informative, a the magnification does not allow easy detection of the nanoparticles in the two-element figure. Similarly, in Figure 5, the fluorescence images appear too small to be easily interpreted, whereas in the separeted files they are much clearer. In the same figure, the SEM panels do not allow a clear observation of the membrane’s status and a higher magnification or an improved focus should be provided.
In the presentation of the results, the section describing nanoparticle synthesis through the different fractions appears somewhat confusing. The authors should consider reorganizing this part to make it clearer, as it is not evident which fraction was ultimately used in the subsequent experiments.
Finally, the Discussion section reads more like a summary or repetition of the results; I would suggest improving it to provide greater interpretative depth and enhance the overall value of the work.
Reviewer 3 Report
Comments and Suggestions for Authors
The authors propose a green extracellular biosynthetic route for Ag₂Se nanoparticles (NPs) using Meyerozyma guilliermondii and explores their potential application as gentamicin adjuvants. While the approach is attractive and relevant, there are notable weaknesses regarding the characterization of biosynthetic conditions and control of biological variability. For instance, it is not specified whether NP production is reproducible across batches or whether key physicochemical properties (size, surface charge, crystallinity) remain stable over time. Furthermore, the influence of free metal ions (Ag⁺, Se²⁻) was not adequately addressed, raising the possibility that the observed antimicrobial effects may stem partly from ionic toxicity rather than Ag₂Se nanoparticles activity.
Although the antibacterial assays employed are diverse (FICI, time–kill kinetics, and biofilm inhibition), the interpretation appears to extend beyond the available experimental evidence. For example, the authors attribute the observed synergy to membrane damage, oxidative stress, and increased permeability, yet no quantitative measurements of intracellular gentamicin accumulation or direct validation of ROS generation using specific inhibitors or probes were included.
The proposed explanation for S. aureus, based on genetic reprogramming, is also limited, as it relies on a small number of genes, lacks independent replication, and provides no evidence that these transcriptional changes are translated at the proteomic or phenotypic level.
The absence of toxicological assessment and evaluation of NP stability under physiological conditions represents another major shortcoming, especially considering the aim of the study. The interaction with serum proteins, as well as potential aggregation or transformation of the nanomaterial in biological fluids, factors that can drastically alter bioactivity and bioavailability, were not analyzed. Possible adverse effects on eukaryotic cells or beneficial microbiota were also not discussed.
minor:
In Figure 1B, the nanoparticles are not clearly visible. See Torres et al., 2012, Journal of Nanoparticle Research 14, 1236 (2012).
In Figure 5, the content presented in Fig(E) through Fig(L) is unclear, and it is difficult to discern the specific features or details the authors intend to highlight.
Overall, this is an excellent and promising piece of work; however, several aspects require clarification and further experimentation. Although the results suggest a strong synergistic effect between Ag₂Se NPs and gentamicin, additional studies focusing on toxicological validation and quantitative analysis of the proposed mechanisms are necessary.
Round 2
Reviewer 2 Report
Comments and Suggestions for Authors
The reviewer thanks the authors for the corrections and appreciate the present form of the paper.
Reviewer 3 Report
Comments and Suggestions for Authors
The review is correct; however, there are some details that need to be corrected. For example, in lines 651 and 687 the scientific names are cut off, and in line 648 there is a grammatical issue